# Genetic Polymorphisms in a Familial Hypercholesterolemia Population from North-Eastern Europe

**DOI:** 10.3390/jpm12030429

**Published:** 2022-03-09

**Authors:** Alexandra Maștaleru, Sabina Alexandra Cojocariu, Andra Oancea, Maria Magdalena Leon Constantin, Mihai Roca, Ioana Mădălina Zota, Irina Abdulan, Cristina Rusu, Roxana Popescu, Lucian Mihai Antoci, Cristian Gabriel Ciobanu, Alexandru Dan Costache, Elena Cojocaru, Florin Mitu

**Affiliations:** 1Department of Medical Specialties I, “Grigore T. Popa” University of Medicine and Pharmacy, 700115 Iaşi, Romania; alexandra.mastaleru@gmail.com (A.M.); leon_mariamagdalena@yahoo.com (M.M.L.C.); roca2m@yahoo.com (M.R.); madalina.chiorescu@gmail.com (I.M.Z.); irina.abdulan@yahoo.com (I.A.); adcostache@yahoo.com (A.D.C.); mitu.florin@yahoo.com (F.M.); 2Clinical Rehabilitation Hospital, 700661 Iaşi, Romania; 3“Saint Spiridon” County Clinical Emergency Hospital, 700111 Iaşi, Romania; 4Department of Medical Genetics, “Grigore T. Popa” University of Medicine and Pharmacy, 700115 Iaşi, Romania; cristina.rusu@umfiasi.ro (C.R.); roxana.popescu2014@gmail.com (R.P.); lucian-mihai.antoci@email.umfiasi.ro (L.M.A.); cristian-gabriel.ciobanu@d.umfiasi.ro (C.G.C.); 5Department of Morphofunctional Sciences I, “Grigore T. Popa” University of Medicine and Pharmacy, 700115 Iaşi, Romania; elena2.cojocaru@umfiasi.ro

**Keywords:** atherosclerosis, metabolic syndrome, ApoB R3500Q, MTHFR C677T, MTHFR A1298C, ACE, PAI-1 4G/5G

## Abstract

(1) Background: Familial hypercholesterolemia (FH) is one of the most prevalent inherited metabolic disorders. The purpose of the study was to investigate the role in cardiovascular disease (CVD) of *PAI-1*, *ACE*, *ApoB-100*, *MTHFR A1298C*, and *C677T*. (2) Methods: From a group of 1499 patients, we included 52 patients diagnosed with FH phenotype and 17 patients in a control group. (3) Results: Most of the FH patients had multiple comorbidities compared to the control group, such as atherosclerosis (48.1% vs. 17.6%), atherosclerotic cardiovascular disease (ASCVD 32.7% vs. 11.8%), and metabolic syndrome (MetS, 40.4% vs. 11.8%). In total, 66.7% of the FH patients had *PAI-1 4G/5G* genotype and MetS. Between *4G/5G* and *4G/4G*, a statistically significant difference was observed (*p* = 0.013). FH patients with *ApoB R3500Q* polymorphism were correlated with ASCVD (*p* = 0.031). Both *MTHFR C677T* and *A1298C* polymorphisms had a significant correlation with gender, alcohol consumption, and smoking status. *ACE* polymorphism was associated with ATS in FH patients, statistically significant differences being observed between heterozygous and homozygous D genotype (*p* = 0.036) as well as between heterozygous and homozygous I genotype (*p* = 0.021). (4) Conclusions: A link between these polymorphisms was demonstrated in the FH group for ATS, ASCVD, and MetS.

## 1. Introduction

Mortality from cardiovascular diseases (CVD) is on the rise, reaching approximately 19.9 million deaths in 2019, 85% of which are caused by coronary artery disease (CAD) and stroke. Globally, it is estimated that by 2030, mortality from CVD will reach approximately 23 million deaths. In the case of surviving a myocardial infarction (MI), there is a high risk of developing complications with frequent hospitalizations and long periods of recovery [1]. Thus, a clear understanding of the pathophysiological mechanisms as well as the factors involved in a premature onset of CVD is necessary to develop reliable screening methods in order to be able to prevent them. CVD are usually determined by the interaction between genetic and environmental factors.

Familial hypercholesterolemia (FH) is defined by very high values of LDL cholesterol that determine in time the development of atherosclerosis process in the arteries (especially in the coronaries and proximal aorta), with subsequent high risk for cardiovascular disease. There are three already well known and widely used clinical diagnostic criteria for FH: MedPed (applied in the United States), Simon Broome (used in the United Kingdom), and Dutch Lipid Clinic Network criteria (DLCN, applied worldwide and most frequently used) [2]. The DLCN score also includes the history of premature CVD for the patient and first degree relatives, high values of LDL (≥190 mg/dL) without any lipid lowering treatment (LLT), and physical signs, such as xanthomata or arcus cornealis [3].

Familial hypercholesterolemia (FH) is a common disorder caused by inherited genetic defects, responsible for high concentrations of low-density lipoproteins (LDL) due to inadequate clearance of the circulating LDL [4,5]. The literature describes a few major defects as being the cause for FH: 87% of the patients have a pathogenic variant in the *LDL-receptor (LDL-R) gene*; 10% in the gene for its ligand, *apolipoprotein B 100 (ApoB-100)*; less than 5% have a defect in the *subtilisin-kexin type 9 proprotein convertase gene (PCSK9)*; and an extremely rare (<1%) cause can be a pathogenic variant in the *LDL-R adaptor protein 1 (LDLRAP1) gene* [6,7]. De Ferranti et al. evaluated in 2016 the incidence of FH as being 1:250 individuals in adults in the USA, but some studies describe even a higher incidence in Danish people. Many authors nowadays consider that this disease has increased substantially in the last two decades, making it one of the most frequent monogenic metabolic diseases in the world [8,9].

The genes involved in FH belong to the network that defines the pathophysiologic mechanism of metabolic syndrome (MetS) and cholesterol metabolism. Changes of these genes could be involved in monogenic or multifactorial forms of FH.

Single nucleotide polymorphisms (SNPs) are common DNA variations usually found between genes that consist in the replacement of a nitrogenous base (adenine—A, thymine—T, cytosine—C, guanine—G) with a different one [10]. Most SNPs have no effect on health or development, but some have been associated with particular responses to certain drugs, susceptibility to environmental factors, or risk to develop specific disorders, such as arterial hypertension (HTA), heart disease, or diabetes mellitus (DM) [11].

Plasminogen activator inhibitor-1 (PAI-1), also known as serpin E1, is the main inhibitor of tissue plasminogen activator and urokinase, both of them activators of plasminogen and hence fibrinolysis. The *PAI-1 gene*, also known as *SERPINE1*, is located on chromosome 7 (7q21.3-q22) and contains a common polymorphism known as *4G/5G* in the promoter region. The *5G allele* is slightly less transcriptionally active than the *4G* [12]. Genotypes *4G/5G* and especially *4G/4G* produce a higher level of *PAI-1* and have been associated with genetic predisposition to MetS, CAD, and stroke [13]. *PAI-1* is increased in certain disorders like obesity, MetS, and some cancers, being involved in the determinism of thrombosis in these conditions [14,15]. *PAI-1* can also induce cellular senescence [16,17], whereas in inflammatory conditions with fibrin deposited in tissues, *PAI-1* appears to play a role in the progression to fibrosis. Angiotensin II increases synthesis of *PAI-1*, so it accelerates the development of atherosclerosis (ATS) [18], *PAI-1* being identified as a predictor for MI [19].

Angiotensin I converting enzyme (ACE) is a carboxypeptidase with a major role in blood pressure regulation and electrolyte balance. It hydrolyses angiotensin I in angiotensin II (potent vasopressor) and aldosterone-stimulating peptide but also inactivates bradykinin (potent vasodilator). Angiotensin II promotes inflammation, reactive oxygen species generation, cell proliferation/apoptosis, fibrosis, and oxidized lipid production [20,21]. The *ACE* gene (located on 17q23.3) contains a polymorphism—an insertion (I) or deletion (D) of a 287 base pair Alu repetitive sequence in intron 16. Being an intronic change, it might have no effects [22], but some recent observational studies and clinical trials have raised the possibility that the *ACE* gene polymorphism modifies the effects of cardiovascular risk factors and their treatment on the risk of CHD [23,24].

Syndrome X, later referred to as MetS, is one of the most intensely studied diseases nowadays, with patients having an increased risk for developing premature CVD and DM. The diagnostic criteria were established by the World Health Organization (WHO) and the National Cholesterol Education Program Expert Panel on Detection, Evaluation, and Treatment of High Blood Cholesterol in Adults (NCEP-ATP III). Thus, MetS is defined by the presence of at least three of the following five criteria: central obesity, glucose intolerance or insulin resistance, dyslipidemia (serum triglycerides (TG) ≥ 150 mg/dL and serum high-density lipoprotein (HDL) ≤ 40 mg/dL), and HTA [25]. The link between PAI-1, MetS, and CVD was demonstrated in many cross-sectional studies [26,27]. Patients evaluated with MetS have a blood hypercoagulability that occurs due to increased levels of clotting and antifibrinolytic factors [28]. Besides its antifibrinolytic role, *PAI-1* also promotes a low-grade inflammation in the system, which is recognized as one of the pathophysiological mechanisms of the onset of MetS. Interleukins, C-reactive protein (CRP), and tumor necrosis factor alpha are some of the proinflammatory factors that stimulate the production of *PAI-1* [29].

Apolipoprotein B is a major component of chylomicrons (ApoB-48), LDL, and VLDL (ApoB-100) and occurs in plasma in two main forms, Apo-B48 (synthesized exclusively by the intestine) and ApoB-100 (synthesized by the liver), both forms being produced by alternative splicing of the same *APOB* premRNA [30]. ApoB-100 is a recognition signal for cellular binding and internalization of LDL particles by the ApoB/E receptor. *APOB* gene is located on 2p24.1, and its pathogenic variants are correlated with FH type 2 and familial hypobetalipoproteinemia, whereas its polymorphisms (e.g., *R3500Q*) are correlated with multifactorial forms of FH [31,32].

Methylenetetrahydrofolate reductase (MTHFR) is an enzyme involved in homocysteine metabolism. The *MTHFR* gene has been mapped to 1p36.3 and may include multiple SNPs, but the most common ones are *C677T* and *A1298C. C677T* polymorphism (exon 4) changes alanine into valine in the catalytic domain, the reason why *TT* homozygotes will have lower enzyme activity and increased homocysteine, leading to cardiovascular effects. *A1298C* polymorphism (exon 7) is located within the enzyme regulatory domain, but the mutant allele (C) affects less protein function [33,34]. 

The aim of the study was to determine the correlations between several gene polymorphisms, certain cardiovascular risk factors, and a series of CVD. Furthermore, the resulting data were compared with a group of patients that have normal lipid values. As a perspective, the prothrombotic status revealed by an adequate genetic test may help identify high-risk individuals with FH who may benefit from primary prevention with an association of antiplatelet medications to conventional LLT that can reduce the cardiovascular risk. 

## 2. Materials and Methods

### 2.1. Study Design and Setting

We performed a cross-sectional study between 1 December 2020–1 December 2021, which was carried out in the Cardiovascular Medical Recovery Clinic in Iasi Clinical Rehabilitation Hospital, an academic medical center in north-eastern Europe.

Patients with a clinical diagnosis of FH based on a DLCN score above 8 were included in the study group. It is considered as a definite FH case if the DLCN score is above 8 points, probable at a score between 6–8, possible at a score between 3–5, and improbable at a score below 3. The value used in the diagnostic score has been adjusted according to the previous LLT administration. Thus, untreated LDL was defined as the absence of LLT in the last 6 months and coincided with the validated laboratory value. In contrast, pre-treated LDL was adjusted to reflect the effects of LLT using a statin-specific correction factor and dose administered alone or in combination with ezetimibe. The correction factors were developed from the analysis of 71 original papers that were collated prior to setting up FH criteria [35].

ASCVD was defined as a history of one of the following diseases: previous MI, acute coronary syndrome, coronary revascularization and other arterial revascularization procedures, stroke and transient ischemic attack, aortic aneurysm, and peripheral artery disease [36].

### 2.2. Study Participants

In a group of 1499 patients hospitalized between 1 December 2020 and 1 December 2021, we identified 52 patients with FH who met the following inclusion criteria: patients who signed informed consent, age over 18 years, and a DLCN score over 8. Patients who did not meet these conditions were excluded from the study. To exclude secondary hypercholesterolemia, any patient with at least one of the following was not included: hypothyroidism, uncontrolled DM, severe liver diseases, severe chronic kidney disease, nephrotic syndrome, and diets elevated in saturated and trans-unsaturated fatty acids. The control group included 17 patients with normal lipid values who signed informed consent and were over 18 years of age. The same exclusion criteria were applied as in the study group (Figure 1). 

### 2.3. Clinical and Biological Evaluation in FH Patients

The anamnesis and the physical examination were performed by a single investigator, the following data being recorded: education, employment and civil status, smoker and drinking status, BMI, hemodynamic status (blood pressure, heart rate), drug history, such as LLT and antiaggregant drugs; and medical history of liver steatosis, ASCVD, HF, MetS, DM, and chronic kidney disease (CKD). Smoker status was characterized in five categories: nonsmoker, previous or occasionally smoking, less than 10 packs year, 11–20 packs year, and more than 20 packs year. Drinking status was defined as nondrinkers, occasional and moderate drinkers (2–20 g alcohol/day), and alcohol abuse (more than 20 g/day).

The laboratory tests performed on the enrolled patients included: TC (mg/dL), LDL (mg/dL), HDL (mg/dL), TG (mg/dL), blood glucose (mg/dL), glycated hemoglobin (HbA1c, %), uric acid (UA, mg/dL), aspartate transaminase (AST/GOT, UI/L), alanine aminotransferase (ALT/GPT, UI/L), CRP (mg/dL), urea (mg/dL), and creatinine (mg/dL). These have been processed using the Erba XL 1000 Spectrophotometer^®^ (Erba Diagnostics, Mannheim, Germany). 

The presence of subclinical atherosclerosis markers such carotid intima-media thickness (IMT) and carotid plaque was used to characterize atherosclerosis (ATS). IMT was performed by a high-resolution B-mode ultrasonography using a single machine (Toshiba Aplio 500) with a linear array 8 MHz scan. IMT was measured between the lumen intima and media-adventitia interfaces of the far wall of the common carotid artery (the 1 cm segment proximal to the bifurcation). Carotid plaque was described as a focal protrusion into the lumen with a thickness of at least 50% larger than the intima-media thickness surrounding it. All the measurements were performed by an experimented investigator.

### 2.4. Genetic Analysis

DNA was isolated from whole blood using PureLink^®^ Genomic DNA Kit (Thermo Fisher Scientific, Waltham, MA, USA) following the protocol supplied by the manufacturer. The presence of the targeted mutations was evaluated with a commercially available kit: CVD StripAssay kit (ViennaLab Diagnostics, Vienna, Austria), which covers 5 mutations: *PAI-1 4G/5G*, *MTHFR C677T*, *MTHFR A1298C*, *ACE I/D*, and *ApoB R3500Q.* The protocol was performed according to the manufacturer’s instructions.

Briefly, 250 ng of genomic DNA was amplified in Sensoquest labcycler gradient (Sensoquest, Göttingen, Germany) using biotinylated primers, in two multiplex reactions. The PCR was performed in a 25 μL reaction volume containing 15 μL amplification mix and 1U of diluted Taq DNA. The program consisted in an initial step of 94 °C for 2 min, followed by 35 cycles of 15 s at 94 °C, 30 s at 58 °C, and 30 s at 72 °C and a final extension step at 72 °C for 3 min.

The last step consisted in hybridization of amplification products to a test strip, which included allele-specific oligonucleotide probes for wild-type and mutant allele, both variants immobilized as an array of parallel lines. Bound sequences were detected after incubation with streptavidin-alkaline phosphatase solution at room temperature for 20 min and color development in the dark, using color substrates containing nitro blue tetrazolium and 5-bromo-4-chloro-3-indolyl phosphate.

Visually positively stained lines were noted in online StripAssay^®^ Online Calculator (ViennaLab Diagnostics, Vienna, Austria, https://www.viennalab.com, accessed on 20 January 2022) and for each targeted sequence were identified as wild-type, heterozygous, or homozygous mutant genotype.

### 2.5. Ethical Approval

All the patients included signed the written informed consent in order to be enrolled in the study. The study was approved by the Ethics Committee of both “Grigore T. Popa” University of Medicine and Pharmacy Iasi (certificate of approval dated 15 June 2020) and Iasi Clinical Rehabilitation Hospital (certificate of approval dated 25 November 2020).

### 2.6. Statistical Methods

Data analysis was performed using SPSS 20.0 (Statistical Package for the Social Sciences, Chicago, Illinois, USA). Data are presented as mean ± standard deviation (SD) or as median with interquartile range for continuous variable and as number of cases with percent frequency for categorical variables. Categorical comparisons were performed by chi-square test or by Fisher’s exact test. Continuous normally distributed variables were compared by Independent Samples *t*-test in case of two samples or by One-Way ANOVA in case of comparisons of more than two samples. The non-parametric Mann–Whitney U test was applied when the distribution of continuous variables did not satisfy the assumption of normality. Analysis of variance (ANCOVA) was applied to adjust for the influence of potential confounders. Correlations between continuous variables were assessed by calculating Spearman correlation coefficients. Multivariate logistic regression analysis was performed to assess several gene polymorphisms as predictors of MetS, ATS, and ASCVD. A two-sided *p*-value < 0.05 was considered significant for all analyses. For the Venn diagrams, we used R program version 4.0.5.

## 3. Results

In our study, we included 52 patients diagnosed with FH and 17 control individuals. The patients included were predominantly females, with a mean age of 55.5 ± 12.8 compared to 38.88 ± 11.72 for males. Most of the patients with FH had multiple comorbidities when compared to the control group, such as atherosclerotic cardiovascular disease (ASCVD, 32.7% vs. 11.8%), heart failure (HF, 19.2% vs. 11.8%), MetS (40.4% vs. 11.8%), and modified basal glycemia (MBG, 48.1% vs. 17.6%). Considering the significant difference in age distribution between case group and control group, we used multivariate analysis to adjust for the influence of this confounder. The general characteristics of the patients included in the study are presented in Table 1, with the non-adjusted results (statistical significance in column with the p-value) as well as age-adjusted results (statistical significance in column *p*-value *) of the comparisons between case group and control group. After age adjustment, significant differences were maintained between FH and control group for the lipidic profile parameters (total cholesterol, LDL cholesterol, corrected LDL cholesterol, non-HDL cholesterol) and for the heart rate.

When applying the DLCN criteria to our groups, 51.9% had a first-degree relative with premature CVD vs. 11.8% in the control group, and 44.2% had a child under 18 years with LDL > 95th percentile vs. 5.9% in the control group. Regarding the personal history of CAD, 32.7% were diagnosed with this disease in the FH group compared to 11.8% in the control group. Most of the patients (42.3%) had an LDL value of more than 330 mg/dL, being scored with 8 points in the DLCN criteria. The descriptive values can be observed in Table 2.

Regarding the *PAI-1 4G/5G* polymorphism (Table 3), from the patients diagnosed with MetS, 66.7% of them had *4G/5G* genotype, followed by 28.6% patients with *5G/5G* polymorphism and 4.7% with *4G/4G* form. A statistically significant correlation was found between the wild-type and heterozygous form in patients diagnosed with MBG. Furthermore, in patients with MetS, a *p* < 0.05 can be observed between wild-type and abnormal homozygous forms. Furthermore, in patients with MetS, there was a statistically significant difference (*p* < 0.05) between the *PAI-1* polymorphism heterozygous and abnormal homozygous genotypes for body mass index (BMI) value. In the control group, the *4G/4G* polymorphism had a mean HDL of 54.57 ± 10.96 mg/dL, and the *5G/5G* had a mean HDL value of 38.66 ± 2.37 mg/dL; it is notable that the *4G/4G* form has a higher HDL value compared to the *5G/5G* form. Significant correlations were observed between the normal homozygous and the heterozygous genotype for the lipid profile (HDL and TG).

Table 4 presents the group of patients with the *ApoB R3500Q* polymorphism, where we identified a significant association between patients with HF and ASCVD (*p* <0.05). Most of the individuals diagnosed with ASCVD had wild-type form (88.2%), and only 11.8% had the heterozygote form. Referring to the patients with homozygous genotype, the mean TC value was 285.47 ± 55.27 mg/dL, and for cLDL, the mean value was 308.24 ± 88.31 mg/dL. In contrast, the patients with the heterozygous form had the mean total cholesterol value 323.50 ± 48.79 mg/dL and LDL mean value of 424.91 ± 142.43 mg/dL.

Patients with the *MTHFR C677T* polymorphism showed a statistically significant difference between the homozygous wild-type and heterozygous form in terms of cardiovascular risk factors: male gender, alcohol consumption, and smoking status. Most of the patients included in our study had a heterozygous form. The patients from this group were 68.4% males and 36.4% females; 75% of these cases were chronic ethanol consumers and 66.7% smokers (Table 5). In addition, a statistically significant difference was observed between the wild-type and abnormal homozygous form in the smoking patients. In the control group, no statistically significant difference was observed between the studied risk factors and the different polymorphisms.

Table 6 presents the same cardiovascular risk factors that were established also for *MTHFR C677T* polymorphism. The *MTHFR A1298C* variant had a statistically positive correlation between the wild-type and abnormal homozygous genotype as well as between the abnormal homozygous and heterozygous when correlated with gender, alcohol consumption, and smoking. The presence of cardiovascular risk factors in addition to the long-term FH can determine the occurrence of heart disease, such as HF, as evidenced by statistical significance with a *p* < 0.05 between normal homozygous and heterozygous compared to the control group. In the control group, there was the same statistically significant difference in terms of smoking status between the normal homozygous and heterozygous form. Another risk factor considered was TC and LDL. Among them, there is a statistically significant difference in patients with normal homozygous genotypes compared to the heterozygous form of *MTHFR A1298C* (*p* < 0.05).

According to Table 7, in FH patients diagnosed with ATS that had *ACE* polymorphism, statistically significant differences were observed between heterozygous and homozygous *D* genotype as well as between heterozygous and homozygous *I* genotype. Most of these patients (60%) had *DI* genotype, while each of the *DD* and *DI* forms represented 20% of the cases. In the control group, statistically significant differences (*p* < 0.05) were observed only in heterozygous and homozygous *D* genotypes for HDL value.

### Multiplicity of Genetic Polymorphisms

As seen in Figure 2a, all patients diagnosed with FH had at least one of the SNPs investigated by us. Participants with at least two, three, or four polymorphisms constituted 94.23% (49), 64.4% (33), and 15.3% (8) of the studied group. None of the patients included in our study were diagnosed with all five polymorphisms. Certain situations of multiplicity predominated over others: 10 (19.2%) patients had polymorphisms in *MTHFR A1298C, ACE*, and *PAI-1* simultaneously, while 8 (15.3%) had polymorphisms in *MTHFR C677T, MTHFR A1298C, PAI-1*, and *ACE* simultaneously. The third case of genetic multiplicity was found in eight (15.3%) patients that had polymorphisms in *MTHFR C677T, PAI-1*, and *ACE* simultaneously.

Figure 2b presents the genetic background of the patients included in the control group. All the patients had at least one of the polymorphisms studied. Most of the participants (94.11%) had at least two associated SNPs, followed by three and four, representing 82.3% and 29.4%, respectively. There were no patients diagnosed with all five studied SNPs. Some situations of multiplicity were more prevalent than others. Similar to the FH group, several scenarios of multiplicity were identified in the control group. Thus, the most frequent simultaneous association was found in five (29.4%) individuals that were diagnosed with *MTHFR C677T, MTHFR A1298C, PAI-1*, and *ACE* polymorphisms, followed by three (17.6%) cases that had polymorphisms in *MTHFR C677T, PAI-1*, and *ACE*. Only two (11.7%) of the study participants had SNPs in *MTHFR A1298C, PAI-1*, and *ACE* concomitantly.

Results of multivariate analysis are presented as odds ratio (OR) with 95% confidence intervals (CIs) (Table 8). Regression analysis identified *PAI 4G/5G* as an independent predictor associated with a 6-fold increase of the risk for MetS. On the contrary, our data proved that ApoB100 wild-type form is an independent protective factor associated with a 30-fold decrease in the risk of developing ASCVD.

## 4. Discussion

The polymorphisms studied in our research focused on ATS, a pathophysiological substrate that underlies the occurrence of CVD, MetS, and ASCVD. We also investigated the possibility of the multiplicity of genetic polymorphisms in these patients, aiming to identify and prevent possible complications. This is the first study that correlates these polymorphisms on an FH population from north-eastern Europe.

In our study, the patients diagnosed with familial hypercholesterolemia were predominantly females (63.5%), the results being consistent with existing data in the literature [37]. A recent study published data that described the fact that male patients are diagnosed earlier with FH compared to women. A possible explanation for this fact can be the multitude of associated risk factors thus determining the earlier diagnosis of the disease [38]. In our group, 18.2% of the females were younger than 44 years, 21.2% were between 45–54 years old, 33.3% were between 55–64 years old, and 27.3% were older than 65 years old. In males, 15.8% were younger than 44 years old, 31.6% were between 45–54 years old, 26.3% were between 55–64 years old, and 26.4% were older than 65 years. Thus, we can report that the males were diagnosed earlier (before age 55) with FH compared to women, as 47.4% of men and 39.4% of women were diagnosed before this limit.

### 4.1. Atherosclerosis

ATS is a complex pathophysiologic process, with a range of early manifestations identified in patients with HF. The present study demonstrated a strong relationship between ATS and ACE, ApoB, and MTHFR genotypes. To date, many studies have reported contradictory results regarding these polymorphisms and atherosclerotic diseases.

Regarding the *ACE* polymorphism, the *D* variant has been associated with an increase of pro-atherosclerotic properties of conventional cardiovascular risk factors [39] as well as worse functional outcome of ischemic stroke, especially in Asians compared to Caucasians [40,41]. Individuals with genotype *II* and *ID* seem to have no cardiovascular consequences, whereas in those with *DD* genotype, the effect of hypercholesterolemia on the risk of CAD seems to be greater [21].

In a study of 610 patients diagnosed with MI compared to a control group of 733 patients, Cambien et al. were the first to report that *ACE DD* genotype is associated with a higher risk of MI both in Caucasians and Asians. The authors found that *ACE DD* variant was associated with a 25% higher risk of CAD compared to *II* genotype [42]. Furthermore, studies have shown that the presence of the *DD* genotype is associated with an increased risk of left ventricular hypertrophy after MI as well as an increased incidence of chronic HF in patients with CAD. Other studies suggest that *DD* genotype combined either with angiotensinogen *TT*, or *GNB3 825* homozygous, or *AT1R* seems to increase the risk for MI, whereas its association with ApoE E4 polymorphism increases the risk for restenosis [22]. A meta-analysis performed on a group of 5169 patients versus 4865 in the control group showed that *DD* genotype is a low-risk factor for CVD in the Chinese Han population [43].

In contradiction, Sayed-Tabatabaei et al. demonstrated that *ACE ID* polymorphism alone is not a strong risk factor for MI, but in young people, its interaction with smoking might influence cardiovascular mortality [44]. Yu Chen et al. conducted a meta-analysis of 40 studies that included 34,993 patients aimed at assessing the link between *ACE ID* and the risk of developing MI. Thus, the study showed that alleles *D* were associated with a 1.41-fold risk for MI [45]. Similar results were demonstrated in another meta-analysis performed on a group of 126,339 patients [46].

According to our study, the polymorphism of the *ACE* gene in patients with FH is positive correlated with ATS, a pathological process that underlies the development of CAD. Our results demonstrated that both homozygous forms of *DD* and *II* are associated with ATS in Caucasian patients in north-eastern Europe, but these results are contrary to Sayed-Tabatabeo’s meta-analysis. One of the reasons that could explain this statistical difference is the discrepancy in the constitution of the groups by including in the study only the Chinese Han subjects. Moreover, it must be taken into account that this population is particular both in terms of genetics and environmental factors, defining elements in the occurrence of ATS.

The second studied polymorphism is *ApoB R3500Q*. Numerous mutations in the *ApoB-100* gene have been described, resulting in a truncation of the protein or amino acid replacement within the protein. These modifications may have an effect on how plasma lipoproteins are metabolized and may therefore have a role in the progression of dyslipidemia [47]. The accumulation in the vascular endothelium of LDL particles enriched with cholesterol and lipoproteins that include ApoB is characteristic of ATS. All main atherogenic lipoproteins contain ApoB, which is an essential protein element [48]. Three mutations in the *APOB-100* gene that determine ApoB-100’s lower affinity for receptors have been described using a molecular genetic technique [47]. 

*ApoB R3500Q* polymorphism may increase CAD risk by elevating LDL levels and coronary artery calcification (CAC). This view is supported by evidence that the *ApoB R3500Q* mutation is a cause of elevated LDL levels, being in a direct relationship between the degree and presence of CAC [49]. Shen et al. reported that for assessing the relationship between CAC and *R3500Q* polymorphism, multiple measurements of LDL are necessary. The explanation offered by the authors was that patients with this polymorphism have had increased levels of LDL since childhood. The authors consider that the *R3500Q* mutation is related to the metabolic pathways of LDL and CAC independent of plasma LDL levels. The study demonstrated that this mutation was not associated with LDL atherogenic more than non-atherogenic LDL. The authors recommend caution in extrapolating the results given the selection bias that may be present [49].

In our study, the *R3500Q* heterozygous variant was associated with ASCVD, with ATS being the underlying cause for CAD. Given the obtained results, we consider that the association would be more noticeable in patients with homozygous mutant variants. The evidence described by [50,51] is not sufficiently well supported without identifying a clear relationship between the presence of the mutation and the early onset of ATS. Therefore, we consider that the changes in the LDL values would cause CAD by calcifying atherosclerotic plaque in patients with the *R3500Q* mutation.

### 4.2. Metabolic Syndrome

Recent studies demonstrated that MetS patients have a high risk for cardiovascular morbi-mortality. However, the traditional risk factors included in the definition of MetS do not explain the burden of vascular pathophysiological changes.

Although the high levels of *PAI-1* are a part of the MetS, it is not included as a criterion for the definition since its evaluation is difficult to perform in routine clinical practice due to the higher costs. Several clinical trials have shown that elevated serum *PAI-1* levels increase the risk of atherosclerotic plaque both by inflammatory pathways and by interfering with cell migration. Contrarily, in obese patients, increased *PAI-1* levels are not associated with inflammation nor dyslipidemia but with visceral obesity and insulin resistance. Studies demonstrated that human visceral adipose tissue synthesizes 5-fold more *PAI-1* than subcutaneous abdominal tissues. Bodyweight loss through diet or comprehensive lifestyle modifications has been associated with reduced plasma levels of *PAI-1*. Furthermore, several studies suggest that circulating *PAI-1* is related to the fat redistribution, not to the fat mass, and it can be used as a biomarker of ectopic fat depots. This emphasizes the role of ectopic fat depots as sites of *PAI-1* synthesis [52].

In addition, some studies suggested that increased *PAI-1* concentrations are a marker for ischemic events in patients with pre-existing ATS, and it may be related to the severity of atherosclerotic lesions. Epidemiological studies have shown a relationship between the rate of acute cardiovascular events and elevated plasma concentrations of *PAI-1.* Scheer FA et al. demonstrated that circadian spikes in *PAI-1* are correlated with the high incidence of MI during the morning [53].

Several researchers described that obesity and MetS were found to be more common in *4G* allele carriers, but this information is contradictory in other studies [54,55,56,57]. Scientists established a link between insulin and pro-insulin levels and the *PAI-1* gene polymorphism 675 *4G/5G* for MI development. Only patients who were homozygous for the 4G allele were at risk for MI with the greatest pro-insulin values, implying that *PAI-1* genotype may alter the vascular risk associated with hyperinsulinemia [19]. Overall, these findings support the idea that *PAI-1* gene diversity plays a role in the modulation of obesity-related phenotypes. According to current studies, the homozygous form of *PAI-1 4G/5G* mutation in the patients with FH included in our study was associated with MetS, BMI, and IFG.

### 4.3. ASCVD

Over time, numerous risk factors for ASCVD have been described, including a family history of CVD, age, sex, sedentary lifestyle, DM or insulin resistance, smoking, overweight or obesity, HTA, and elevated blood cholesterol levels, especially LDL [58]. A gene that has implications in the atherosclerosis process is *MTHFR*, two of the most studied polymorphisms being *C677T* and *A1298C*. In our study, both *MTHFR* polymorphisms correlated with gender, alcohol, and smoking status, all of them being risk factors for ATS.

The association between *MTHFR C677T* polymorphism and increased risk of developing CVD has been demonstrated in a series of studies. For example, Chao Xuan et al. performed a meta-analysis that included 30 case-control studies involving a group of 8140 patients with both *MTHFR C677T* polymorphism and MI compared to a group of 10,522 controls. The meta-analysis included subjects of different ethnicities, so there were twenty-one studies on Caucasian individuals, four studies on East America, and one study from Africa-America and aimed to investigate the link between *MTHFR C677T* polymorphism and MI. The authors reported that *MTHFR C677T* polymorphism was associated with a risk of MI in Caucasian patients under 50 years, but there was no link between elderly patients of any other ethnicity [59]. Another meta-analysis of 14 studies involving 2981 Chinese Has individuals identified the relationship between *MTHFR C677T* polymorphisms and the risk of developing CAD. Furthermore, the author demonstrated that subjects with *TT* genotype are more likely to develop CAD compared to the *CC* genotype carriers. This emphasizes that the *T* allele is prone to CAD [60]. On the other hand, there are several studies that have not identified an association between *MTHFR C677T* polymorphism and CAD. A meta-analysis of 31 studies covering 2780 cases and 3022 controls aimed at assessing the risk of CVD in patients with *MTHFR C677T* in a Turkish population did not find a significant association between CAD and the polymorphism. The authors consider that the differences are mainly caused by dietary habits and folate acid deficiency [61].

Regarding the *MTHFR A1298C* polymorphism and CAD, the data in the literature are controversial. A case-control study was performed to determine the prevalence of *MTHFR* polymorphism, *C677T*, and *A1298C*. The authors concluded that both homozygote and heterozygote *A1298C* carriers are associated with a significantly higher risk of developing early CAD [62]. Contrarily, a study of a Tunisian population of 352 patients diagnosed with CAD compared to a group of 390 healthy subjects, age and gender-matched, showed that homozygosity of *MTHFR C677T* polymorphism was linked with CAD but not *A1298C*. Ghazouani et al. described the genetic differences of the two polymorphisms and their association with CAD. It is possible that the functional differences in the correlation of the *MTHFR* polymorphisms with CAD are related to the different locations of the two SNPs. The *C677T* polymorphism is found in exon 4 inside the catalytic domain of the *MTHFR* gene, whereas the *A1298C* polymorphism, located in exon 7, is placed within the enzyme regulatory domain of the gene [63].

### 4.4. Multiplicity of Genetic Polymorphisms

The current study demonstrated that the most frequent three or four concomitant associated polymorphisms found in both groups were *MTHFR C677T, MTHFR A1298C, PAI-1,* and *ACE*. AlBancha et al. evaluated patients with hypercholesterolemia and HTA, comparing them with patients without these two comorbidities. The study showed that 32.6% of the subjects included in the study group had five mutations. The comparison between the study group with the control group showed that 27–33% of those included in the study had 4–5 mutations simultaneously, these being predominant in the group of hypertensive and dyslipidemic patients. These mutations were the same as those identified by us, highlighting the issue of an active screening at the populational level both in terms of risk factors and genetic evaluation in order to prevent major cardiovascular events [64].

Recently, Damar and Eroz studied both genetic and risk factors that could increase the chance of developing MI. The authors observed a statistically significant higher presence of *MTHFR A1298C, PAI-1,* and *ACE* polymorphisms in smoking patients who had a MI compared to the patients in the control group. In our study population, three of these genes were observed simultaneously in 19.2% of the patients diagnosed with FH. Due to the fact that this category of patients has an increased risk of developing MI both from FH and genetic evaluation, they requested a personalized primary prevention through regular medical evaluation, lifestyle changes, and regular exercise [65].

## 5. Conclusions

In conclusion, the current research evaluated the role in CVD of a series of genetic polymorphisms, such as *PAI-1, ACE, ApoB-100, MTHFR A1298C*, and *C677T*, respectively, among the FH population compared to a control group. The results from our study demonstrated an association between these polymorphisms identified in a FH group and ATS, ASCVD, and MetS, together increasing the cardiovascular risk. Our results regarding the presence of several polymorphisms in patients with cardiovascular comorbidities emphasize the data from the literature that a comprehensive evaluation is mandatory in HF patients. Thus, prevention would be the key in monitoring patients with HF and implicitly would lead to a decrease in long-term morbidity and mortality. Both the FH and control groups had the same genetic polymorphisms multiplicity but in different percentages, with more than half of the patients included in the study having two or three associated polymorphisms. However, this is the first study to evaluate these gene polymorphisms in a HF population from north-eastern Europe. In order to have a better understanding of this association, large studies are needed, focusing on gene–environment interaction.

## 6. Limitations

We performed a unicentric study that included patients admitted to Clinical Rehabilitation Hospital from Iasi, which covers the population from north-east Romania. Given the fact that the study was developed during the COVID-19 pandemic period, the number of patients who underwent clinical medical evaluation was limited (larger number of patients diagnosed with SARS-CoV-2 infection, including medical staff). Therefore, the incidence of the disease was reduced. The high prevalence of women compared to males in our study could be explained by the high level of anxiety experienced by women facing the pandemic period—women were more careful about their health status, whereas men avoided medical evaluation. In addition, as there were differences in the characteristics of the control group compared to the study group in terms of glomerular filtration rate, we considered this to be a limitation of the study, and we did not make correlations in terms of CKD.

## Figures and Tables

**Figure 1 jpm-12-00429-f001:**
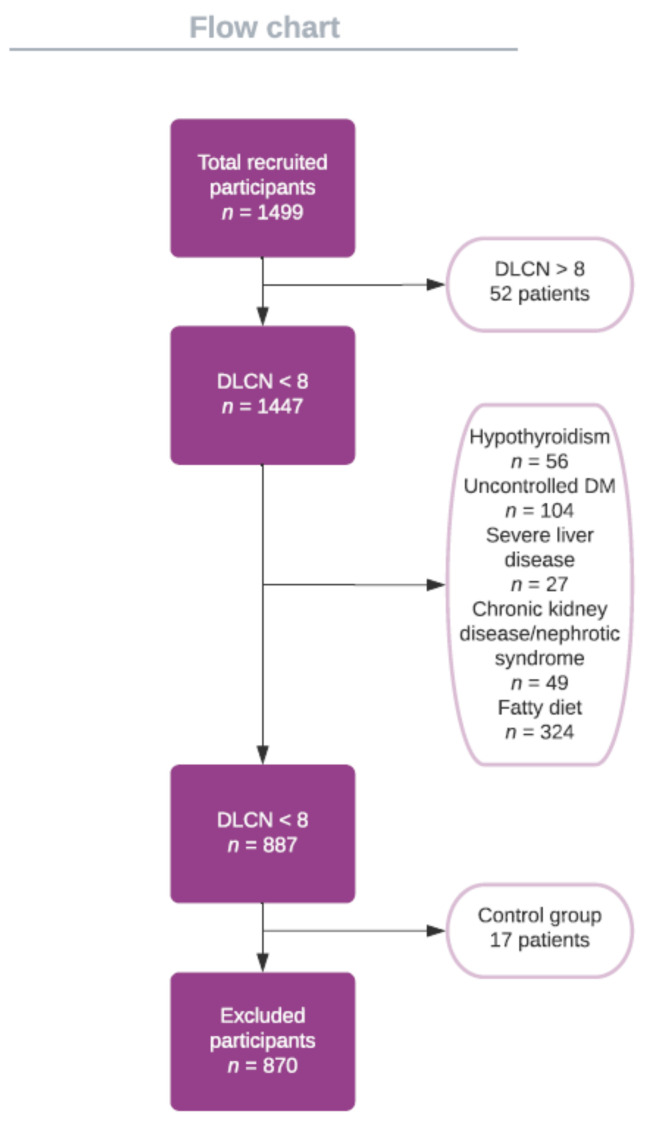
Flow chart of the studied group.

**Figure 2 jpm-12-00429-f002:**
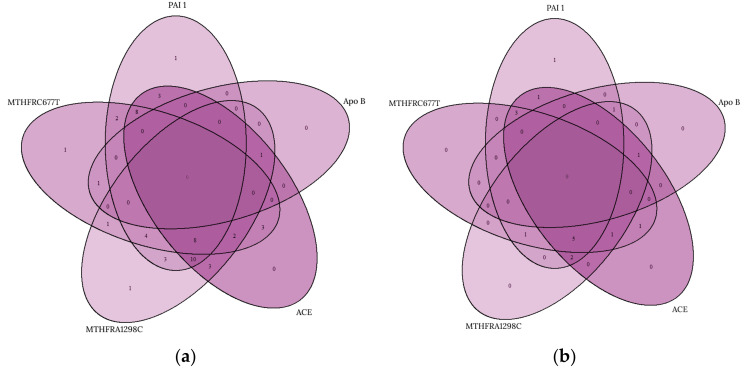
Multiplicity of the studied polymorphisms in the FH group (**a**) and control group (**b**).

**Table 1 jpm-12-00429-t001:** General characteristics of the studied population.

General Characteristics		FH	Controls	*p*-Value	*p*-Value *
Gender, n (%)					
	Males	19 (36.5)	6 (35.3)	0.927	0.563
	Females	33 (63.5)	11 (64.7)		
Age, years (mean ± SD)		55.5 ± 12.80	38.88 ± 11.72	0.000	-
Education, n (%)					
	8 classes	5 (9.6)	2 (11.8)	0.607	0.725
	12 classes	18 (34.6)	2 (11.8)		
	University	20 (38.5)	12 (70.5)		
	Other	9 (17.3)	1 (5.9)		
Employment status, n (%)					
	Full-time employed	24 (46.2)	13 (76.5)	0.032	0.773
	Part-time employed	1 (1.9)	1 (5.9)		
	Retired	24 (46.2)	2 (11.8)		
	Unemployed	3 (5.8)	1 (5.9)		
Civil status, n (%)					
	Not married	2 (3.8)	5 (29.4)	0.027	0.725
	Married	41 (78.8)	10 (58.8)		
	Divorced	4 (7.7)	1 (5.9)		
	Widow/er	5 (9.6)	1 (5.9)		
Smoker status, n (%)					
	Non smoker	31 (59.6)	10 (58.8)	0.881	0.544
	Ex/occasionally smoking	4 (7.7)	1 (5.9)		
	Less than 10 packs year	6 (11.5)	4 (23.5)		
	11–20 packs year	5 (9.6)	1 (5.9)		
	More than 20 packs year	6 (11.5)	1 (5.9)		
Drinking status, n (%)					
	Non drinkers	28 (53.8)	9 (52.9)	0.950	0.645
	Occasionally drinkers	21 (40.4)	7 (41.2)		
	Alcohol abuse	3 (5.8)	1 (5.9)		
BMI, kg/m^2^ (mean ± SD)		28.71 ± 5.22	23.99 ± 5.89	0.003	0.375
Hemodynamic status					
	SBP, mmHg (mean ± SD)	125.53 ± 10.10	124.05 ± 6.59	0.575	0.624
	DBP, mmHg (mean ± SD)	72.46 ± 7.56	71.88 ± 3.85	0.764	0.944
	MBP, mmHg (mean ± SD)	88.01 ± 8.41	85.58 ± 5.44	0.269	0.714
	PP, mean ± SD	53.07 ± 6.89	52.17 ± 5.67	0.628	0.436
	HR, beats/min (mean ± SD)	67.57 ± 7.39	73.88 ± 4.15	0.001	0.050
Lipidic profile, mg/dL (mean ± SD)					
	Total cholesterol	286.93 ± 55.10	207.24 ± 42.15	0.000	0.000
	HDL cholesterol	58.52 ± 17.51	46.34 ± 11.12	0.009	0.084
	Non-HDL cholesterol	228.40 ± 57.52	160.90 ± 43.39	0.000	0.003
	LDL cholesterol	202.41 ± 45.86	129.62 ± 40.05	0.000	0.000
	Corrected LDL cholesterol	312.73 ± 91.67	129.62 ± 40.05	0.000	0.000
	Triglycerides	185.88 ± 120.05	178.07 ± 276.96	0.871	0.464
Glucose, mg/dL (mean ± SD)		128.34 ± 144.65	98.64 ± 8.25	0.403	0.838
Uric acid, mg/dL (mean ± SD)		4.44 ± 1.57	4.45 ± 1.37	0.971	0.248
CRP, mg/dL (mean ± SD)		0.91 ± 1.56	0.62 ± 0.59	0.452	0.305
Lipid-lowering therapy, n (%)					
	No lipid-lowering therapy	15 (28.8)	17 (100)	-	-
	Statin	35 (67.4)	0 (0)		
	Statin + ezetimib	2 (3.8)	0 (0)		
Antiaggregant drugs, n (%)					
	COX-inhibitor	14 (26.9)	1 (5.9)	0.070	0.342
	P2Y12 receptor blockers	5 (9.6)	0 (0)	-	-
Hepatic steatosis, n (%)		23 (44.2)	2 (11.8)	0.016	0.394
ATS, n (%)		25 (48.1)	3 (17.6)	0.045	0.619
ASCVD, n (%)		17 (32.7)	2 (11.8)	0.096	0.686
HF, n (%)		10 (19.2)	2 (11.8)	0.024	1.000
Metabolic syndrome, n (%)		21 (40.4)	2 (11.8)	0.039	0.479
MBG, n (%)		25 (48.1)	3 (17.6)	0.045	0.200
T2DM, n (%)		6 (11.5)	0 (0)	-	-
CKD, n (%)					
	Stage 1	12 (23.1)	8 (47.1)	0.009	1.000
	Stage 2	25 (48.1)	9 (52.9)		
	Stage 3A	13 (25.0)	0 (0)		
	Stage 3B	2 (3.8)	0 (0)		

FH, familial hypercholesterolemia; BMI, body mass index; SBP, systolic blood pressure; DBP, diastolic blood pressure; MBP, mean blood pressure; PP, pulse pressure; HR, heart rate; HDL, high-density lipoprotein; LDL, low-density lipoprotein; CRP, C- reactive protein; ASCVD, atherosclerotic cardiovascular disease; HF, heart failure; MBG, modified basal glucose; T2DM, type 2 diabetes mellitus; CKD, chronic kidney disease; *p*-value, statistical significance between FH group and control group; *p*-value *, age-adjusted statistical significance between FH group and control group.

**Table 2 jpm-12-00429-t002:** Dutch Lipid Clinic Network diagnostic criteria applied to the studied population.

*DLCN criteria*	FH	Controls	*p*-Value
First-degree relative			
Premature CVD	27 (51.9%)	2 (11.8%)	0.004
LDL > 95th percentile	15 (28.8%)	3 (17.6%)	0.528
Tendinous xantoma a/o arcus cornealis	3 (5.8%)	1 (5.9%)	0.986
Child with LDL > 95th percentile	23 (44.2%)	1 (5.9%)	0.003
Personal history			
Premature CVA/PVD	6 (11.5%)	0 (0%)	-
Coronary artery disease	17 (32.7%)	2 (11.8%)	0.124
Physical examination			
Arcus cornealis < 45 years of age	16 (30.8%)	1 (5.9%)	0.040
LDL			
155–189 mg/dL	0 (0%)	7 (41.2%)	-
190–249 mg/dL	15 (28.8%)	0 (0%)	-
250–329 mg/dL	15 (28.8%)	0 (0%)	-
>330 mg/dL	22 (42.3%)	0 (0%)	-

FH, familial hypercholesterolemia; CVD, cardiovascular disease; LDL, low-density lipoprotein; CVA, cerebrovascular accident; PVD, peripheral vascular disease.

**Table 3 jpm-12-00429-t003:** Correlation between *PAI 1 4G/5G* polymorphism and metabolic anomalies.

	5G/5G	4G/5G	4G/4G	P_1_	P_2_	P_3_
*PAI-1*						
TC	273.73 ± 57.54	298.98 ± 52.32	274.11 ± 57.02	0.390	1.000	0.418
LDL	185.71 ± 42.16	211.89 ± 44.25	199.15 ± 51.21	0.187	0.758	0.739
c-LDL	315.17 ± 103.89	331.07 ± 83.14	268.82 ± 88.98	0.880	0.465	0.125
HDL	61.56 ± 21.05	56.94 ± 18.11	58.80 ± 12.09	0.777	0.914	0.924
non-HDL	212.16 ± 47.74	242.04 ± 60.60	215.30 ± 56.72	0.225	0.988	0.394
TG	190.16 ± 142.61	189.78 ± 114.64	172.47 ± 115.38	1.000	0.938	0.902
IFG	3 (12%)	17 (68%)	5 (20%)	0.041 *	0.411	0.299
BMI	28.87 ± 4.67	30.15 ± 5.30	25.29 ± 4.26	0.719	0.134	0.014 *
MetS	6 (28.6%)	14 (66.7%)	1 (4.7%)	1.000	0.035 *	0.013 *
*PAI-1* control						
TC	194.46 ± 19.88	201.54 ± 53.42	218.42 ±38.56	0.950	0.441	0.781
LDL or c-LDL	138.03 ± 19.79	111.27 ± 47.65	144.37 ± 34.54	0.457	0.930	0.334
HDL	38.66 ± 2.37	41.41 ± 8.42	54.57 ± 10.96	0.718	0.019 *	0.067
non-HDL	155.80 ± 20.36	160.12 ± 58.95	163.85 ± 37.21	0.984	0.900	0.989
TG	88.73 ± 3.98	281.52 ± 427.43	112.90 ± 20.64	0.499	0.049 *	0.580
IFG	0 (0%)	1 (33.3%)	2 (66.7%)	-	-	1.000
BMI	23.00 ± 3.39	24.29 ± 6.13	24.11 ± 7.11	0.908	0.941	0.999
MetS	0 (0%)	1 (50%)	1 (50%)	-	-	

TC, total cholesterol; LDL, low-density lipoprotein; c-LDL, corrected LDL; HDL, low-density lipoprotein; TG, triglycerides; IFG, impaired fasting glucose; BMI, body mass index; MetS, metabolic syndrome; P1, statistical significance between wild type and heterozygous form; P2, statistical significance between wild-type and homozygous form; P3, statistical significance between heterozygote and homozygote form; * *p* < 0.05.

**Table 4 jpm-12-00429-t004:** Association between *ApoB R3500Q* polymorphism with ASCVD and lipid profile.

	Wild Type	Heterozygote	*p*
*ApoB*			
TC	285.47 ± 55.27	323.50 ± 48.79	0.344
LDL	202.26 ± 45.93	206.20 ± 62.36	0.907
c-LDL	308.24 ± 88.31	424.91 ± 142.43	0.077
HDL	58.84 ± 17.03	50.60 ± 36.06	0.519
non-HDL	226.62 ± 57.93	272.90 ± 12.72	0.269
TG	176.72 ± 104.37	414.90 ± 303.49	0.466
ASCVD	15 (88.2%)	2 (11.8%)	0.031 *
*ApoB*—control			
TC	210.05 ± 43.95	186.20 ± 19.51	0.470
LDL/c-LDL	129.68 ± 42.55	129.15 ± 17.60	0.986
HDL	47.22 ± 11.57	39.75 ± 2.05	0.389
non-HDL	162.82 ± 45.79	146.45 ± 17.46	0.632
TG	190.28 ± 293.78	86.45 ± 0.63	0.634
ASCVD	2 (100%)	0 (0%)	-

TC, total cholesterol; LDL, low-density lipoprotein; c-LDL, corrected LDL; HDL, low-density lipoprotein; TG, triglycerides; ASCVD, atherosclerotic cardiovascular disease; * *p* < 0.05.

**Table 5 jpm-12-00429-t005:** *MTHFR C677T* and cardiovascular risk factors.

	Wild Type	Heterozygote	Homozygote	P_1_	P_2_	P_3_
*MTHFR C677T*						
Gender (F)	18 (54.5%)	12 (36.4%)	3 (9.1%)	0.032 *	0.303	1.000
Gender (M)	4 (21.1%)	13 (68.4%)	2 (10.5%)			
Alcohol	4 (16.7%)	18 (75%)	2 (8.3%)	0.000 *	0.303	0.300
Smoking	5 (23.8)	14 (66.7)	2 (9.5)	0.040 *	0.018 *	0.489
*MTHFR C677T*—control						
Gender (F)	2 (18.2%)	7 (63.6%)	2 (18.2%)	0.136	-	-
Gender (M)	4 (66.7%)	2 (33.3%)	0 (0%)			
Alcohol	4 (50%)	4 (50%)	0 (0%)	0.386	-	-
Smoking	4 (57.14%)	3 (42.86%)	0 (0%)	0.333	-	-

F, female; M, male; P1, statistical significance between wild-type and heterozygous form; P2, statistical significance between wild type and homozygous form; P3, statistical significance between heterozygote and homozygote form; * *p* < 0.05.

**Table 6 jpm-12-00429-t006:** *MTHFR A1298C* and cardiovascular risk factors.

	Wild Type	Heterozygote	Homozygote	P_1_	P_2_	P_3_
*MTHFR A1298C*						
Gender (F)	10 (30.3%)	12 (36.4%)	11 (33.3%)	1.000	0.046 *	0.038 *
Gender (M)	9 (47.4%)	9 (47.4%)	1 (5.3%)			
Alcohol	11 (45.8%)	11 (45.8%)	2 (8.4%)	0.761	0.032 *	0.043 *
Smoking	11 (52.4%)	7 (33.3%)	3 (14.3%)	0.142	0.025 *	0.014 *
HF	7 (70%)	2 (20%)	1 (10%)	0.049 *	0.108	0.693
TC	292.84 ± 50.33	283.55 ± 59.36	283.49 ± 58.55	0.855	0.892	1.000
LDL	202.82 ± 49.29	200.94 ± 43.78	204.31 ± 47.77	0.991	0.996	0.978
C-LDL	295.46 ± 72.90	309.74 ± 92.34	345.30±114.34	0.850	0.390	0.635
HbA1c	6.10 ± 1.59	5.30 ± 0.47	6.11 ± 1.37	0.112	1.000	0.161
*MTHFR A1298C*—control						
Gender (F)	4 (36.4%)	6 (54.5%)	1 (9.1%)	1.000	1.000	1.000
Gender (M)	2 (33.3%)	3 (50%)	1 (16.7%)			
Alcohol	3 (37.5%)	5 (62.5%)	0 (0%)	0.386	-	-
Smoking	4 (57.1%)	2 (28.6%)	1 (14.3%)	0.036 *	0.108	0.163
HF	0 (0%)	2 (100%)	0 (0%)	-	-	-
TC	174.40 ± 26.66	211.22 ± 24.91	287.90 ± 16.82	0.053	0.013 *	0.054
LDL	112.28 ± 20.80	147.14 ± 26.25	102.80 ± 109.46	0.035 *	0.992	0.858
C-LDL	112.28 ± 20.80	147.14 ± 26.25	102.80 ± 109.46	0.035 *	0.992	0.858
HbA1c	5.03 ± 0.13	5.35 ± 0.39	5.24 ± 0.007	0.111	0.029 *	0.693

F, female; M, male; HF, heart failure; TC, total cholesterol; LDL, low-density lipoprotein; c-LDL, corrected LDL; HbA1c, glycated hemoglobin; P1, statistical significance between wild type and heterozygous form; P2, statistical significance between wild type and homozygous form; P3, statistical significance between heterozygote and homozygote form; * *p* < 0.05.

**Table 7 jpm-12-00429-t007:** Relationship between *ACE DI* polymorphism with lipid profile and atherosclerosis.

	D/D	I/I	D/I	P_1_	P_2_	P_3_
*ACE*						
TC	293.78 ± 30.93	264.61 ± 64.92	300.42 ± 55.69	0.249	0.894	0.186
LDL	207.94 ± 34.25	190.35 ± 52.06	208.48 ± 47.43	0.506	0.999	0.515
C-LDL	321.39 ± 90.22	295.29 ± 91.96	321.07 ± 94.81	0.709	1.000	0.677
HDL	61.15 ± 18.67	55.81 ± 17.57	58.98 ± 17.24	0.698	0.936	0.843
non-HDL	232.63 ± 33.91	208.80 ± 62.42	241.44 ± 63.67	0.381	0.857	0.264
TG	196.40 ± 167.09	167.76 ± 67.21	193.53 ± 120.71	0.821	0.998	0.686
ATS	5 (20%)	5 (20%)	15 (60%)	1.000	0.036 *	0.021 *
*ACE*—control						
TC	237.00 ± 41.72	194.50 ± 68.58	201.93 ± 39.22	0.756	0.479	0.988
LDL/LDL-C	157.33 ± 32.68	127.35 ± 49.00	123.07 ± 40.78	0.765	0.372	0.993
HDL	57.43 ± 4.98	54.90 ± 15.27	42.15 ± 9.39	0.973	0.018 *	0.692
non-HDL	179.56 ± 36.87	139.60 ± 53.31	159.78 ± 45.33	0.690	0.729	0.881
TG	116.00 ± 12.99	97.65 ± 75.02	206.99 ± 328.48	0.941	0.618	0.591
ATS	1 (33.3%)	1 (33.3%)	1 (33.3%)	-	-	-

TC, total cholesterol; LDL, low-density lipoprotein; c-LDL, corrected LDL; HDL, low-density lipoprotein; TG, triglycerides; ATS, atherosclerosis; P1, statistical significance between *D/D* and *I/I* form; P2, statistical significance between *D/D* and heterozygote form; P3, statistical significance between heterozygote and *I/I* form; * *p* < 0.05.

**Table 8 jpm-12-00429-t008:** Multivariate regression analysis to predict relationship of *PAI1* and *ApoB* gene polymorphism with MetS and ASCVD, respectively.

	Odds Ratio	95% CI	*p*-Value
MetS–*PAI 4G/5G*	14.43	2.32–89.65	0.004
ASCVD–*ApoB100* wild type	0.029	0.001–0.929	0.045

## Data Availability

Not applicable.

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
