# Peer review of "Genetic Polymorphisms in a Familial Hypercholesterolemia Population from North-Eastern Europe"

_jpm, 2022, doi:10.3390/jpm12030429_

Round 1

Reviewer 1 Report

This is well written manuscript, descriptive and informative.  However, there are several issues that need to be addressed.

Major:

  1. Clearly the sample size is the major limitation of the study. The validity of all statistical analyses performed in the manuscript is doubted. It needs to be addressed. 
  2. Aside from the sample size issue, one of main confounders might to be the difference in age distribution between case group and control group. Age adjusted tests are needed.
  3. From Table 1, all control subjects are either stage 1 or 2 CKD patients. This needs to be addressed.
  4. Given the overall sample size limitation already, it is even harder to draw any conclusion from further subgrouping done in “Multiplicity of genetic polymorphism” section. Instead, it might be better to do an aggregation approach like GRS analysis.

Minor:

  1. In lines 71-72, “either pathogenic variants (leading to monogenic forms of FH) or polymorphisms (involved in multifactorial forms of FH)”, the use of the terms ‘pathogenic’ vs. ‘polymorphism’ is confusing. All disease causing genes are pathogenic. A genetic variant is either ‘monomorphic’ or ‘polymorphic’.  A disease/trait is either ‘monogenic’ or ‘polygenic’.
  2. In line 228, “The patients included were predominantly females”, this is interesting. Is this a known sex difference for FH or is it for the current study only? If so, is there a plausible explanation?  
  3. In line 241, “for age and sex”, this seems to be a misplaced.
  4. In line 270, “cLDL” not “LDL”?
  5. In line 289, only MTHFR A1298C is presented in Table 6, not for “both” MTHFR studied polymorphisms.

Author Response

Dear reviewer,

Thank you for all your comments.

  1. We performed a unicentric study that included patients admitted to the Clinical Rehabilitation Hospital from Iasi, which deserves the population from the N-E Romania. Due to the low addressability during the pandemic and the reduced possibility of enrolling patients (large number of covid patients, including medical staff), as well as the low incidence of the disease, the number of patients included in the study was limited.
  2. Given that Romania is a country with a very high cardiovascular risk according to the 2021 ESC Guidelines on cardiovascular disease prevention in clinical practice, the prevalence of normal lipid profile values ​​after 40 years is low. We have added a column in Table 1 that includes the p values ​​adjusted for age of all the patients characteristics. Statistically significant values were observed ​​(p <0.05) for almost all the lipid profile.
  3. As there were differences in the characteristics of the control group compared to the study group in terms of glomerular filtration rate, we considered this to be a limitation of the study and we did not make correlations in terms of CKD.
  4. Thank you for your recommendation. We do not have any experience with GRS analysis but we will learn about it and use it in further articles. Instead, we have realized a linear regression on multiplicity and we have added it in the text.
  5. Familial hypercholesterolemia could be inherited as a monogenic disorder (the defect of the LDL receptor with autosomal dominant inheritance) or as a multifactorial disorder (produced by many DNA defects + unfavourable action of the environment).

Monogenic disorders are produced by pathogenic gene variants (initially called mutations). The actual classification of gene variants (mutations) that is used worldwide identifies 5 classes: pathogenic / probably pathogenic / VUS (variant of unknown significance) / probably benign / benign. For monogenic disorders the DNA change (mutation) is located in the coding part of the gene and the function of the gene involved is usually moderately/severely affected.

Polymorphisms are very small DNA changes (a single nucleotide involved) that eventually alter mildly the function of the gene. They could be located in the coding or noncoding part of the gene. Most polymorphisms have no role in health/disease, but some have been identified as predisposing factors in multifactorial disorders. Each polymorphism individually has a small influence on the risk to develop a multifactorial disorder, but multiple polymorphisms together could produce a significant risk. This is what we wanted to explore in the actual study, knowing that such studies are very limited in the field of familial hypercholesterolemia.

  1. Many studies and meta-analyses described the fact that familial hypercholesterolemia is more frequent in women compared to men, but males are diagnosed earlier due to the associated risk factors such as smoking or alcohol consumption. In addition, as the study developped during the COVID 19 pandemic period, we have included all the patients that addressed for medical evaluation and accomplished the selection criteria. A plausible explanation of the selection bias could refer to the psychologic profile of the person facing a pandemic – women worry more about their own health, whereas men try to avoid medical centres.
  2. We have modified the rest of the minor issues. Thank you for your remark!

Hope we have touched all the points you asked us to change.

If there are any other changes you consider we should make, please let us know.

Yours sincerely,

All the authors

Reviewer 2 Report

Mastaleru and colleagues studied the genetic polymorphisms of PAI-1, ACE, ApoB-100, and MTHFR in patients clinically diagnosed with familial hypercholesterolemia. These polymorphisms have been known to play a role in the development of cardiovascular diseases in the normal population as well in several special groups of patients, however, they have been less studied in familial hypercholesterolemia.

Major:

  1. Study design: n of control group is very low (less than 1/3 of FH group). In several cases even categorical variables of subgroups have been analyzed (table 1: metabolic syndrome n=0 vs n=1 vs n=1; table 2: ASCVD 2 vs 0; table 7: ATS 1 vs 1 vs 1), where there is an absolute lack of statistical power. The corresponding FH results cannot be compared to them this way.
  2. The FH and control groups are not age-matched. Control group is significantly younger, which has a very high impact on the results, especially in case of cardiovascular diseases.
  3. The authors report multiplicity of the studied polymorphisms. Since this can have a major confounding effect, regression analysis would have been the appropriate statistical analysis.
  4. ApoB100 R3500Q polymorphism has been analyzed as an independent factor in FH patients. As it is also mentioned in the introduction (line 60-61), ApoB100 mutations can have a causative effect in patients with a clinical diagnosis of FH. One of the known mutations is ApoB100 R3500Q itself (https://doi.org/10.1038/ncpcardio0836). In my opinion, it should not be analyzed as an independent factor if FH is a phenotypic diagnosis.
  5. One of the main results highlighted in the abstract is following: “FH patients with ApoB R3500Q polymorphism were correlated with ASCVD (p =0.031)”. This sentence can be misleading, since the majority (88.2%) of FH patients diagnosed with ASCVD had a wild type genotype (table 4). On the other hand: What was the frequency of ApoB100 R3500Q polymorphism in the whole group? It would be interesting to see whether it is similar in the whole FH group or only in patients with ASCVD.
  6. A similar question is raised about the results showed in table 7. What is the distribution of D/D, I/I, and D/I genotypes in the whole group? Is it special in patients with ATS or it just reached significance because of the n?
  7. The diagnosis of ATS is not specified in the methods.

Minor:

  1. PAI-1 is not identically written (line 26-27 PAI 4G/5G; line 261 PAI 1)
  2. There is a formatting error in subtitle 2.2 and 2.3.
  3. Typo: 429.91 (line 272) versus table 4: 424.91. Please check which value is correct.

Author Response

Dear reviewer,

Thank you for all your comments.

  1. We have looked over all the tables again and changed where there was a lack of statistical power. Thank you!
  2. Given that Romania is a country with a very high cardiovascular risk according to the 2021 ESC Guidelines on cardiovascular disease prevention in clinical practice, the prevalence of normal lipid profile values ​​after 40 years is low. We have added a column in Table 1 that includes the p values ​​adjusted for age of all the patients characteristics. Statistically significant values were observed ​​(p <0.05) for almost all the lipid profile.
  3. Using the multivariate regression analysis, we have predicted the relationship of PAI1 and ApoB gene polymorphism with the metabolic syndrome and ASCVD. We have introduced in the article a table that describes this relationship. Thank you for your advice!
  4. ApoB100 R3500Q is a polymorphism (minor change of the gene), not a proper mutation (pathogenic gene variant). We have studied polymorphisms that are considered to be involved in cardiovascular risk, to see if they are valid for multifactorial forms of familial hypercholesterolemia. As in any multifactorial disorder, some genetic defects are more important (and we expect this should be a key element, as the gene is directly involved in cholesterol metabolism), whereas others are less important (we expect the rest of the polymorphisms have a smaller contribution, as they are involved due to the pathogenic mechanism).
  5. At your suggestion, we calculated the percentage of patients with ApoB100 R3500Q polymorphism in the whole group of FH, which was 3.8%. We considered it useful to mention in the article the percentage for the group of patients diagnosed with HF and ASCVD.
  6. Comparing with the data written in table 7, in which the distribution of D/D, I/I, and D/I genotypes were 20%, 20% and 60%, in the whole group the percentages were 26.9, 32.7 and 40.4, meaning that it is significant in patients with ATS, not because of n.
  7. Thank you for your remark. We have added in the Materials and Methods section the diagnosis of ATS.
  8. Regarding the minor comments, we have modified them. Thank you!

Hope we have touched all the points you asked us to change.

If there are any other changes you consider we should make, please let us know.

Yours sincerely,

All the authors

Round 2

Reviewer 1 Report

1. This limitation is not addressed in the manuscript.

3. This limitation is not addressed in the manuscript.

4. Is the result of the newly added 'multivariate analysis' from FH case group only or combined?  It's not clear what value this new result is adding regard to the main trait of interest in this manuscript, i.e., FH.   Any confounders corrected?   Any multicollinearity issue and how is it dealt with? 

5.  There are more than one way to classify the genetic variants in different academic discipline.  Regardless, using the term 'polymorphism' analogous to SNP is not appropriate when SNP is a type of polymorphism.  Moreover, the 'polymorphism' is not a contrasting concept to 'pathogenicity' of genetic variant.   Therefore, the phrase “either pathogenic variants or polymorphisms" does not make sense.

6. Author's note written in the response to reviewers make some sense, but not the way written in the manuscript.  In your data set, do you also see more females in the older FH patients than in younger patients?  

Author Response

Dear reviewer,

Thank you for all your comments.

  1. We have added a Limitation section in the article.
  2. The newly added results represent the comparison between the two studied groups, these being age-adjusted. Through multivariate analysis, we have eliminated the confounding factors. After this adjustment, the differences for the lipid profile and heart rate remained significant, these variables (lipid profile) being of interest to us.
  3. We have changed in the text the phrase. Thank you!
  4. We have added in the manuscript more details regarding the age of diagnosis for males and females. Yes, we have more older females in our data set. That was a very good idea. Thank you!

Hope we have touched all the points you asked us to change.

If there are any other changes you consider we should make, please let us know.

Yours sincerely,

All the authors

Reviewer 2 Report

The authors have addressed my previous concerns and significantly improved the manuscript.

Author Response

Dear reviewer, 

Thank you again for all your remarks! They will help us in our future research.

Yours sincerely,

All the authors